# Enhancing Cybersecurity in Healthcare: Evaluating Ensemble Learning Models for Intrusion Detection in the Internet of Medical Things

**DOI:** 10.3390/s24185937

**Published:** 2024-09-13

**Authors:** Theyab Alsolami, Bader Alsharif, Mohammad Ilyas

**Affiliations:** 1Department of Electrical Engineering and Computer Science, Florida Atlantic University, 777 Glades Road, Boca Raton, FL 33431, USA; talsolami2021@fau.edu (T.A.); ilyas@fau.edu (M.I.); 2College of Computer, Najran University, Najran 61441, Saudi Arabia; 3Department of Computer Science and Engineering, College of Telecommunication and Information, Technical and Vocational Training Corporation, Riyadh 12464, Saudi Arabia

**Keywords:** healthcare, security threats in IoMT, machine learning, intrusion detection system

## Abstract

This study investigates the efficacy of machine learning models for intrusion detection in the Internet of Medical Things, aiming to enhance cybersecurity defenses and protect sensitive healthcare data. The analysis focuses on evaluating the performance of ensemble learning algorithms, specifically Stacking, Bagging, and Boosting, using Random Forest and Support Vector Machines as base models on the WUSTL-EHMS-2020 dataset. Through a comprehensive examination of performance metrics such as accuracy, precision, recall, and F1-score, Stacking demonstrates exceptional accuracy and reliability in detecting and classifying cyber attack incidents with an accuracy rate of 98.88%. Bagging is ranked second, with an accuracy rate of 97.83%, while Boosting yielded the lowest accuracy rate of 88.68%.

## 1. Introduction

The Internet of Medical Things (IoMT) is transforming the healthcare industry by establishing connections between medical devices, applications, and healthcare IT systems. This interconnected ecosystem enables real-time patient monitoring, personalized treatment plans, and improved healthcare outcomes. Devices such as wearable health monitors, smart infusion pumps, and connected imaging machines provide continuous data, facilitating proactive healthcare management. These advancements contribute to more accurate diagnoses, timely interventions, and tailored care strategies, ultimately enhancing patient health outcomes and quality of life [1].

Despite these benefits, the integration of IoMT devices into healthcare networks presents significant cybersecurity challenges. These devices, often designed with limited computational power and security features, are vulnerable to various cyber threats, including unauthorized access, data breaches, and malware infections. The implications of such vulnerabilities are severe, as they can lead to unauthorized access to sensitive patient information, manipulation of device functionality, and disruption of healthcare services. For instance, a compromised insulin pump could administer incorrect dosages, posing life-threatening risks to patients. These security issues not only jeopardize patient data privacy but also undermine the safety and reliability of medical services [2,3,4].

Current security measures, including traditional IDSs, are increasingly inadequate in addressing the complex and evolving threats targeting IoMT devices. Traditional IDSs, which rely on predefined signatures and rule-based detection, struggle to recognize new or sophisticated attack patterns. Furthermore, the vast and dynamic nature of IoMT networks exacerbates these limitations, as the volume and diversity of connected devices create a broader attack surface that is challenging to monitor and secure effectively [1,2].

Given these inadequacies, there is a critical need for advanced security solutions that can adapt to the evolving threat landscape of IoMT. Machine learning-enhanced IDS offers a promising approach by continuously analyzing network traffic and device behavior to detect anomalies and potential threats in real-time. By leveraging machine learning, IDS can learn from new attack patterns and adapt to emerging threats, providing solid and dynamic protection for IoMT devices [5].

Machine learning algorithms used in IDS can be trained on large datasets of network traffic and device behavior to recognize patterns associated with both normal operations and malicious activities. These algorithms can detect previously unknown threats, making IDS more effective than traditional security measures. Moreover, machine learning enables IDS to adapt to new types of attacks, ensuring ongoing protection as cyber threats evolve [4,5].

This research focuses on the vulnerability of IoMT devices to unauthorized access and data breaches, investigating the use of ensemble learning techniques to enhance the detection and classification of attacks in IoMT networks, thereby protecting patient data and ensuring the reliability of healthcare services.

The key contributions of this research paper can be outlined as follows:We designed a scheme based on machine learning technology to enhance the classification of attacks versus normal operations.We applied feature selection methods to improve the IDS performance of IoMT network devices.We used ensemble learning techniques, including Stacking, Bagging, and Boosting.We evaluated the performance of our scheme in terms of accuracy, precision, recall, and F1-score.Our scheme outperformed recent studies utilizing the same dataset, specifically in Stacking and Bagging.

This paper is organized as follows:

Section 2 outlines the background information regarding IoMT connections and associated security threats. Section 3 reviews the relevant literature and examines various machine learning methodologies employed for safeguarding the IoMT. Section 4 elaborates on the study’s methodology, detailing the approaches and techniques implemented. Section 5 presents the results and discusses the application of ensemble learning techniques. Finally, Section 6 provides a comprehensive conclusion.

## 2. Background

This section provides a comprehensive overview of IoMT data flow, challenges, and security considerations essential for advancing healthcare through connected technologies and underscores their implications for developing effective IDS solutions using machine learning algorithms.

### 2.1. Internet of Medical Things (IoMT) Connection Overview

IoMT involves a network of interconnected medical devices, sensors, and software applications that communicate and exchange data to improve patient care and streamline healthcare processes [6]. The following is an overview of how IoMT devices send and receive information through networks:

#### 2.1.1. Data Collection

IoMT devices, including wearable health monitors, smart implants, and connected medical equipment, continuously gather data from patients. These data can encompass vital signs (e.g., heart rate, blood pressure), glucose levels, activity levels, and other health metrics [2].

#### 2.1.2. Local Processing

Some IoMT devices have local processing capabilities that allow them to perform a preliminary analysis of the collected data. For instance, a wearable heart monitor might detect abnormal heart rhythms and generate alerts without needing to send raw data to a central server [7].

#### 2.1.3. Data Transmission

IoMT devices employ diverse communication protocols and network technologies for transmitting data to other devices or central servers [2,8]. The key technologies include:Wi-Fi: many IoMT devices utilize local Wi-Fi networks to transmit data to healthcare providers or cloud servers.Bluetooth: wearable IoMT devices commonly utilize Bluetooth to transmit data to smartphones or gateways, which then forward the data to central systems.Cellular Networks: certain IoMT devices, particularly those for remote monitoring, integrate cellular connectivity to transmit data directly over mobile networks.Zigbee/Z-Wave: low-power wireless protocols such as Zigbee and Z-Wave are employed in some IoMT devices to facilitate short-range communication within healthcare facilities.

#### 2.1.4. Data Aggregation and Storage

The transmitted data are typically aggregated and stored in centralized systems, such as cloud-based servers or healthcare provider databases. These systems provide secure storage, ensuring that patient data are protected and compliant with healthcare regulations [9].

#### 2.1.5. Data Processing and Analysis

Central servers or cloud platforms perform advanced data processing and analysis on the aggregated data [7,9]. This analysis can include:Real-Time Monitoring: continuously monitoring patient data to detect anomalies or critical events, triggering alerts for healthcare providers.Predictive Analytics: leveraging machine learning algorithms to anticipate potential health problems based on past and current data.Data Integration: combining data from multiple IoMT devices and other sources (e.g., electronic health records) to provide comprehensive insights into a patient’s health.

#### 2.1.6. Communication and Feedback

The processed data and analysis results are communicated back to healthcare providers, caregivers, and patients [10]. This feedback loop can involve:Alerts and Notifications: sending real-time alerts to healthcare providers when critical thresholds are crossed (e.g., a significant drop in blood oxygen levels).Reports and Dashboards: providing healthcare providers with detailed reports and dashboards to monitor patient health trends.Patient Feedback: sending notifications and recommendations to patients via mobile apps or other devices, encouraging them to take specific actions (e.g., medication reminders, and exercise prompts).

#### 2.1.7. Secure Communication

Throughout this entire process, ensuring secure communication is paramount [7,10]. IoMT devices and networks use various security measures, including:Encryption: encrypting data during transmission to protect it from interception and unauthorized access.Authentication: verifying the identity of devices and users to prevent unauthorized access to the network.Access Control: implementing strict access control policies to ensure that only authorized personnel can access sensitive patient data.Regular Updates: keeping device firmware and software up-to-date to protect against known vulnerabilities.

Refer to Figure 1 for the data flow within the IoMT.

### 2.2. Challenges and Security Threats in IoMT

The proliferation of the IoMT presents a transformative opportunity to enhance patient care through interconnected medical devices, sensors, and software applications. However, the integration of IoMT into healthcare systems introduces a range of challenges and security threats that must be meticulously addressed to ensure the safety, privacy, and reliability of medical data and devices.

### 2.3. Challenges in IoMT

This section provides a detailed examination of several key challenges and considerations in the IoMT.

#### 2.3.1. Interoperability

The IoMT comprises devices from various manufacturers, each employing distinct communication protocols and standards. This diversity necessitates the development of robust interoperability frameworks to ensure seamless data exchange and integration across heterogeneous systems. Achieving interoperability is critical for enabling comprehensive patient monitoring and coordinated healthcare delivery [7,11].

#### 2.3.2. Data Management

IoMT devices generate vast amounts of data continuously. Effective data management strategies are essential to store, process, and analyze these data efficiently. The challenges include ensuring data quality, managing storage capacities, and facilitating real-time data processing to support timely clinical decision-making [7,11].

#### 2.3.3. Regulatory Compliance

Compliance with stringent healthcare regulations, such as the Health Insurance Portability and Accountability Act (HIPAA), is paramount in IoMT systems. These regulations mandate the protection of patient data privacy and security [12]. IoMT solutions must be designed and implemented to meet these regulatory requirements, which involves adopting rigorous data protection measures and conducting regular compliance audits [7,11].

#### 2.3.4. Scalability

As the adoption of IoMT devices grows, healthcare systems must scale to accommodate the increasing volume of data and the number of connected devices. Scalability challenges encompass network infrastructure, data processing capabilities, and system performance. Scalable IoMT solutions are necessary to maintain reliable and uninterrupted healthcare services [13].

### 2.4. Security Threats in IoMT

This section discusses various security threats and vulnerabilities faced by the IoMT devices and systems:

#### 2.4.1. Data Breaches

IoMT devices collect and transmit sensitive patient information, making them attractive targets for cyberattacks. Unauthorized access to these data can lead to significant privacy breaches, exposing patients to identity theft and other malicious activities. Ensuring data confidentiality and integrity is a critical security concern in IoMT environments [14].

#### 2.4.2. Malware and Ransomware

IoMT devices are susceptible to malware and ransomware attacks, which can disrupt healthcare services and compromise patient safety. Malware infections can alter device functionality, leading to inaccurate data collection and erroneous clinical decisions. Ransomware attacks can lock down critical medical systems, demanding ransom payments to restore access [7,14,15].

#### 2.4.3. Device Hijacking

Cybercriminals can hijack IoMT devices to exploit their computational resources for malicious purposes, such as launching distributed denial-of-service (DDoS) attacks. Device hijacking can also involve tampering with medical treatments, posing direct risks to patient health and safety [15,16].

#### 2.4.4. Insider Threats

Insider threats, originating from within healthcare organizations, represent a significant security risk. These threats can arise from malicious intent or negligence by employees, contractors, or other trusted individuals. Insider threats can lead to data breaches, unauthorized access, and intentional or unintentional damage to IoMT systems [7,14,15].

#### 2.4.5. Physical Attacks

Physical security is a fundamental aspect of protecting IoMT devices. Physical tampering or theft of devices can compromise their functionality and security. Ensuring physical security measures, such as secure device enclosures and tamper-evident features, is essential to prevent unauthorized access and manipulation [17].

### 2.5. The Role of IDS in IoMT Security

#### 2.5.1. Anomaly Detection

Machine learning-enhanced IDSs (ML-IDSs) play a pivotal role in securing IoMT environments by monitoring network traffic for anomalies that may indicate security breaches or malicious activities [18]. IDSs leverage advanced algorithms to detect deviations from typical behavior, facilitating the prompt identification of potential threats [17,19].

#### 2.5.2. Real-Time Alerts

ML-IDSs provide real-time alerts to healthcare providers and IT security teams, facilitating prompt responses to detected security incidents. Real-time alerts enable the rapid mitigation of threats, minimizing potential damage and ensuring the continued safety and integrity of IoMT systems [20].

#### 2.5.3. Enhanced Visibility

Continuous monitoring of network traffic by IDSs enhances visibility into IoMT environments. This visibility allows for the identification of vulnerabilities, assessment of security postures, and implementation of proactive security measures to address emerging threats [16,20].

## 3. Related Work

Various studies have been conducted to address the pressing need for robust ML-IDSs in the IoMT to combat cyber threats and safeguard sensitive healthcare data. Despite significant advancements, challenges persist in handling high-dimensional data, managing resource constraints, and ensuring real-time threat detection.

The research by [21] focused on employing an ensemble learning stacking method to detect cyberattacks and protect the IoMT. Their system, which utilized Random Forest, Gradient Boosting, and Support Vector Machine (SVM) as base models, achieved a 96.9% accuracy rate using the WUSTL EHMS 2020 dataset. However, the study did not address the challenge of high data dimensionality, which can lead to overfitting and decreased accuracy.

The authors in [22] described an ensemble learning framework that enhances IoT device security through binary classification of normal and abnormal traffic, employing models like Random Forest and Logistic Regression. While the study achieved a high accuracy rate of 98.64% using the TON-IoT dataset, it primarily focused on general IoT devices and did not delve into the specific challenges of medical IoT environments, such as real-time processing and handling sensitive biometric data.

Reference [23] tackled some of the challenges by proposing a novel feature selection technique, Logistic Redundancy Coefficient Gradual Upweighting MIFS (LRGU-MIFS), which improved accuracy in the IoMT by effectively reducing dimensionality. Although this method demonstrated significant improvements over traditional approaches, there remains room for optimizing model efficiency and ensuring real-time deployment in resource-constrained environments.

The development of Intelligent and Explainable IDSs by [24] introduced an innovative approach combining Particle Swarm Optimization (PSO) for feature engineering with SHapley Additive exPlanations (SHAP) for model interpretability. This method enhanced transparency and accuracy, achieving a 96.56% accuracy rate, but did not explicitly address the integration of such systems in highly dynamic and heterogeneous IoMT networks.

The model proposed by [25] effectively integrates the Random Forest algorithm with an advanced feature scaling technique to handle large and complex categorical data, particularly within the context of IoMT networks. This approach is particularly advantageous for e-healthcare systems that require efficient processing of extensive datasets. By reducing feature dimensions and the number of instances, the framework significantly enhances classification speed while maintaining a high level of accuracy, achieving an average accuracy of 94.23%.

In [26], the authors proposed a Federated Bayesian Optimization XGBoost model for detecting cyberattacks in IoMT systems. To enhance the model’s effectiveness, they employed feature selection by eliminating all irrelevant features from the dataset. However, in their experiments, the authors utilized raw data without implementing any advanced techniques for further improving the model’s performance.

Our research proposes a novel hybrid approach that integrates advanced feature selection techniques with ensemble learning, specifically tailored for IoMT environments. This approach not only improves detection accuracy by mitigating overfitting and managing high-dimensional data but also enhances real-time processing capabilities through optimized computational efficiency. This research contributes to the field by providing a scalable and efficient IDS framework that can be deployed in resource-constrained IoMT networks using machine learning algorithms, potentially setting new standards for security in healthcare IoT applications.

As shown in Table 1, our proposed scheme compares favorably with recent studies in terms of dataset type, attack types, accuracy, and methodology.

## 4. Methodology

Building an Intrusion Detection System (IDS) using machine learning algorithms involves several essential phases to ensure robust and accurate detection of network and device anomalies. The process begins with acquiring a certified dataset that includes both network flow metrics and patient biometrics, which are crucial for addressing security challenges within the Internet of Medical Things (IoMT). This dataset serves as the foundation for the IDS, providing the necessary information to distinguish between normal operations and potential attacks.

Next, data preprocessing is imperative to achieve optimal accuracy rates. This step includes tasks such as feature scaling, normalization, and dimensionality reduction, which prepare the data for effective model training and evaluation.

The selection of appropriate machine learning algorithms is guided by critical parameters, including data characteristics, types of attacks, scalability, accuracy, and performance metrics. For the proposed IDS, an ensemble learning technique has been implemented, utilizing Random Forest and Support Vector Machine (SVM) as base models. Random Forest excels in handling high-dimensional data and mitigating overfitting through its ensemble structure, while an SVM is adept at handling complex patterns and high-dimensional spaces with its ability to find optimal decision boundaries.

The performance of this ensemble approach is compared across different methodologies—stacking, boosting, and bagging. Stacking combines predictions from multiple base models using a meta-model to improve accuracy, boosting sequentially trains weak learners to correct errors made by previous models, and bagging reduces variance by aggregating predictions from multiple instances of the same base algorithm. By evaluating these techniques, the IDS aims to identify the most effective strategy for robust and accurate real-time attack detection.

### 4.1. Dataset Description

The WUSTL-EHMS-2020 dataset was developed using a real-time Enhanced Healthcare Monitoring System (EHMS) testbed, which integrates network flow metrics and patient biometrics—a novel offering in current dataset collections [27]. This testbed includes medical sensors, a gateway, network infrastructure, and visualization controls, enabling data to flow from patient sensors through the gateway to a server for visualization via switches and routers. However, this data flow is susceptible to interception by malicious actors, potentially compromising data integrity [27].

The dataset captures instances of man-in-the-middle attacks, including spoofing and data injection. Spoofing involves unauthorized packet sniffing, which jeopardizes the confidentiality of patient data, while data injection refers to on-the-fly modifications of packets during transit, undermining data integrity [27]. The network flow metrics and patient biometric data are recorded in CSV format using the Audit Record Generation and Utilization System (ARGUS) tool. The dataset comprises 44 features: 35 network flow metrics, 8 biometric features, and 1 label feature, where samples linked to attacker MAC addresses are marked as 1, and others are marked as 0 [27]. Figure 2 illustrates the dataset’s statistical attributes.

To prepare the dataset for analysis, we conducted cleaning, normalization, and balancing. A widely accepted approach is to allocate 80% of the data for training and validation, reserving the remaining 20% for testing. This split is recommended in both academic and practical contexts, ensuring sufficient data for model training while maintaining a robust assessment of model performance on unseen data. We employed the train–test–split function from the scikit-learn library to randomly partition the dataset, specifying a random state to ensure reproducibility. This method maintains a consistent split across different runs, facilitating reliable comparisons in model performance.

By utilizing this random split technique, we ensured that both training and testing sets are representative of the overall dataset, which is essential for developing robust machine learning models. This approach not only aids in effective model evaluation but also minimizes the risk of overfitting, enhancing the models’ generalization capabilities in real-world scenarios. Overall, this systematic approach to dataset splitting maximizes the potential of the WUSTL-EHMS-2020 dataset, ultimately improving intrusion detection and safeguarding patient data within healthcare systems.

### 4.2. Data Preprocessing

Effective intrusion detection in the IoMT hinges upon thorough data preparation. Cleaning processes ensure dataset integrity by eliminating duplicates and managing missing values. Normalization techniques standardize numerical features, enabling equitable comparisons and enhancing model performance. Balancing methodologies address class imbalance, thereby improving IDS sensitivity to both attack and normal instances. These foundational steps are essential for developing reliable IoMT security solutions, as detailed in the subsequent subsection.

#### 4.2.1. Data Cleaning

Data cleaning is a crucial step in preparing the dataset for analysis, ensuring data integrity and reliability. In our study, the dataset underwent meticulous cleaning procedures. Initially, duplicate records were identified and removed by comparing rows across key columns to ensure the uniqueness of each data point. This step was essential to prevent bias that could arise from redundant data entries. Additionally, comprehensive checks for missing values across all columns were conducted. Where possible, missing values were handled through imputation using statistical measures such as the mean or median. This meticulous approach not only enhances the overall quality of the dataset but also lays a solid foundation for subsequent analysis and modeling tasks, ensuring that insights derived from the data are accurate [28].

#### 4.2.2. Data Normalization

Normalization is essential to standardize the scale of numerical features within a dataset, facilitating fair comparisons and improving the performance of machine learning models. In our study, we employed min–max scaling to normalize the numerical attributes of the dataset. This process rescales each feature to a range between 0 and 1, preserving the relative relationships between data points while mitigating the impact of varying scales [29].
(1)$$X_{\mathrm{normalized}}=\frac{X-\min(X)}{\max(X)-\min(X)}$$

In this context, $X_{\mathrm{normalized}}$ denotes the scaled value, X refers to the original data point, min(X) indicates the minimum value of the feature within the dataset and max (X) represents the maximum value of the feature within the dataset [30].

Min–max scaling was preferred over other normalization methods such as Z-score normalization or robust scaling because it maintains the original distribution of data while ensuring that all features are within the same range. This is particularly advantageous in scenarios like using an IDS for IoMT, where maintaining the relative importance of features is crucial for accurately detecting anomalies.

By normalizing features such as network flow metrics and biometric data, we ensure that each attribute contributes equally to model training and evaluation. This standardized approach not only enhances the model’s convergence during training but also promotes better interpretability of results, ultimately supporting more accurate insights into healthcare data security and anomaly detection within IoMT environments.

#### 4.2.3. Data Balancing

In the domain of machine learning, especially within the context of IDS, addressing class imbalance is crucial for enhancing model performance and reliability. Class imbalance occurs when the number of instances of one class significantly outnumbers those of the other class, leading to a biased model that favors the majority class. This can be particularly problematic in IDS, where detecting attacks (minority class) is critical.

To address the class imbalance in our dataset, we employed the Random Over-Sampling technique, a method that involves duplicating samples from the minority class to equalize the class distribution. The process was initiated by segregating the features (X) from the target variable (y) [31]. Following this, the RandomOverSampler was applied, resulting in a resampled dataset where both the attack and normal classes contained an equal number of instances. This approach ensures that the machine learning model receives balanced exposure to both classes during training, thus mitigating bias towards the majority class and enhancing the model’s capability to accurately detect the minority class, which is crucial for the reliability and effectiveness of the IoMT IDS. Figure 3 shows the balanced dataset.

### 4.3. Data Selection

Mutual Information (MI) and Principal Component Analysis (PCA) have been employed to select and process features from the dataset for training machine learning algorithms aimed at safeguarding the IoMT using an IDS. MI is a statistical measure that quantifies the dependency between variables, crucially assessing how much information about one variable (such as attack labels in the IoMT) can be inferred from another (each feature in the dataset) [23]. This approach is ideal for IoMT IDSs because it identifies features most informative for detecting anomalies or attacks in IoMT environments.

The rationale for utilizing MI in this context lies in its ability to capture both linear and non-linear relationships between features and attack behaviors without assuming a specific data distribution. Unlike linear methods, which may overlook complex dependencies, MI ensures that critical patterns indicative of attacks in IoMT systems are effectively identified. This flexibility is particularly advantageous in IoMT applications where the nature of attacks can vary widely. The mathematical formulation of Mutual Information (MI) is as follows:(2)$$I\left(X;Y\right)=\sum_{y\in Y}\sum_{x\in X}p\left(x,y\right)\log\left(\frac{p(x)p(y)}{p(x,y)}\right)$$
where:*X* and *Y* are two random variables.p(x,y) is the joint probability mass (or density) function of *X* and *Y*.p(x) and p(y) are the marginal probability mass (or density) functions of *X* and *Y*, respectively.

However, MI is not without its limitations. One significant drawback is its sensitivity to noise in the data. Since MI quantifies dependencies between variables, noisy or irrelevant features can distort these dependencies, leading to suboptimal feature selection. To mitigate this issue, several strategies were employed. First, data preprocessing steps, including noise filtering and feature engineering, were applied to minimize the impact of irrelevant or misleading data points. Additionally, careful feature selection was conducted to ensure that only the most informative features were retained, reducing the likelihood of noise affecting the final model. Following the feature selection using MI, PCA is applied to the selected features to further reduce dimensionality while preserving the most significant variance. PCA transforms the selected features into a set of principal components, which capture the maximum variance in the data. This combined approach enhances the performance of IDS models by focusing on the most informative features and then reducing dimensionality for computational efficiency.

By selecting features based on MI and then applying a PCA, the goal is to improve IDS model performance through a more manageable feature set that retains critical information and reduces redundancy. This process not only improves computational efficiency but also enhances model interpretability by prioritizing features with the highest predictive power. Therefore, MI-based feature selection combined with PCA aligns with the objective of optimizing IoMT IDS performance, ensuring robust detection capabilities across diverse attack scenarios. Figure 4 shows the top 10 features selected based on Mutual Information, while Table 2 provides a detailed description of these features.

### 4.4. Random Forest

Random Forest, a versatile ensemble learning method in machine learning, has garnered significant attention for its efficacy in IDS. IDS play a crucial role in cybersecurity by identifying unauthorized access and potential threats to computer networks and systems. In this context, Random Forests excel due to their ability to handle high-dimensional data, complex feature interactions, and inherent noise typically present in network traffic data.

One of the key strengths of Random Forests is their ensemble nature, where multiple decision trees are independently trained on different subsets of the data and then aggregated to make predictions. This ensemble approach not only improves prediction accuracy but also enhances the model’s robustness against overfitting—a common challenge in IDS where classifiers must generalize well to new and unseen attack patterns. However, while Random Forests are designed to mitigate overfitting through their ensemble structure, it is important to recognize that their generalization effectiveness can vary based on the diversity and representativeness of the training data. The risk of overfitting to specific attack patterns or network environments remains a concern, especially when the training data does not adequately cover the full spectrum of potential threats.

In practice, Random Forests in IDS are trained on labeled datasets containing historical network traffic data, where each data point is labeled as normal or malicious. During training, the model learns to distinguish between normal network behavior and various types of attacks, such as denial-of-service (DoS), intrusion attempts, and malware activities. This learning process enables the classifier to detect anomalies and suspicious patterns in real-time network traffic, thereby preemptively alerting administrators to potential security breaches.

To optimize the model’s performance, careful feature selection and dimensionality reduction steps are employed, such as using Mutual Information (MI) to select the top features and applying Principal Component Analysis (PCA) to reduce dimensionality. Additionally, the RandomizedSearchCV technique is used for hyperparameter tuning, further enhancing the model’s ability to generalize by focusing on the most informative features and reducing the risk of overfitting. Random Forests are adept at capturing nonlinear relationships and interactions between features, which is essential in IDS, where malicious activities often manifest in complex and dynamic ways. By combining multiple decision trees, each trained on a random subset of features and data samples, Random Forests can effectively model diverse attack scenarios and adapt to evolving threat landscapes [32].

The interpretability of Random Forests enhances their utility in IDS applications. By analyzing feature importance scores derived from the ensemble of decision trees, security analysts can gain insights into the most influential network features indicative of malicious activity. This interpretability not only aids in understanding the underlying factors contributing to detected threats but also facilitates informed decision-making in deploying appropriate mitigation strategies.

### 4.5. Support Vector Machine

Support Vector Machine (SVM) is a powerful supervised machine learning algorithm that has gained significant popularity in various fields due to its effectiveness in classification tasks [29,33]. SVM is particularly well-suited for IDSs in the realm of cybersecurity. An IDS is a critical component in network security that helps identify and respond to potential security threats and malicious activities. SVM offers several advantages that make it highly useful in the context of IDSs.

One of the main advantages of SVM is its ability to handle high-dimensional data efficiently and effectively. Network traffic data in IDS often consist of numerous features or attributes that characterize the behavior of network traffic. SVM excels in handling such high-dimensional data and identifying complex patterns within it, which is crucial for distinguishing between normal and anomalous network traffic patterns. Additionally, SVM is known for its ability to find the optimal hyperplane that separates different classes in the feature space while maximizing the margin between them. This capability enables robust and accurate classification of network traffic into normal and malicious categories, thereby facilitating the detection of various cyber threats and attacks.

Furthermore, SVM is inherently strong against overfitting, which is vital in IDS, where the model needs to generalize well to unseen data and adapt to evolving cyber threats. By finding the optimal decision boundary with the maximum margin, SVM achieves good generalization performance, maintaining high accuracy in detecting anomalies in network traffic. The flexibility of SVM in supporting different kernel functions—such as linear, polynomial, and radial basis function (RBF) kernels—enables it to handle non-linear relationships in the data. This adaptability is particularly beneficial in IDS, where the patterns of malicious activities may not be linearly separable in the feature space.

SVM supports different kernel functions, such as linear, polynomial, and radial basis function (RBF) kernels, which enables the algorithm to handle non-linear relationships in the data. This flexibility in modeling complex relationships in the data is particularly beneficial in IDS where the patterns of malicious activities may not be linearly separable in the feature space [34].

### 4.6. Ensemble Learning

Ensemble learning has emerged as a powerful paradigm in machine learning, leveraging the synergy of multiple models to enhance predictive performance beyond what individual models can achieve [22,33]. This approach capitalizes on the diversity and complementary strengths of constituent models, thereby mitigating weaknesses and improving overall robustness. Three prominent techniques within ensemble learning are stacking, boosting, and bagging, each contributing uniquely to the ensemble’s effectiveness.

#### 4.6.1. Stacking

Stacking involves a two-level approach where a meta-model, such as Logistic Regression, is trained on the predictions or probabilities outputted by base models, including Random Forests and Support Vector Machines (SVMs). This meta-model consolidates the predictions from various base models to produce a final, more accurate prediction. The theoretical advantage of stacking lies in its ability to combine the strengths of diverse models, addressing their individual weaknesses and leveraging their collective insights to handle complex data patterns more effectively [22,33]. Stacking was chosen over methods like simple averaging or voting due to its capability to learn optimal weights for combining base model predictions, thus offering a more refined and potentially higher-performing result.

#### 4.6.2. Boosting

Boosting sequentially trains a series of weak learners to correct errors made by preceding models. Each subsequent learner focuses on instances where previous models struggled, thereby iteratively improving predictive accuracy. Boosting algorithms, such as AdaBoost and Gradient Boosting, are known for their ability to handle class imbalance and emphasize challenging instances, resulting in more refined predictions. Each subsequent model in the boosting process aims to correct the errors made by the previous models, improving the overall prediction accuracy [22,33].

#### 4.6.3. Bagging

Bagging, or Bootstrap Aggregating, involves training multiple instances of the same base algorithm on different subsets of the training data, sampled with a replacement. By aggregating the predictions through averaging or voting, bagging reduces variance and increases model stability, which is particularly beneficial for unstable learners like decision trees. Random Forests are a notable example of bagging, where multiple decision trees are trained on bootstrapped samples and their collective predictions are averaged to enhance generalization and robustness [22,33]. The choice of bagging and Random Forests is justified by their effectiveness in reducing overfitting and improving stability, attributes that are advantageous compared to models with higher variance or sensitivity to noise.

The selection of Random Forests and SVM as base models in these ensemble methods is based on their distinct strengths. Random Forests provide robustness through multiple decision trees and are adept at handling large datasets with high-dimensional features. SVMs, on the other hand, are known for their effectiveness in high-dimensional spaces and their ability to find optimal decision boundaries. By incorporating these models into the ensemble methods, the approach capitalizes on their complementary strengths, thereby enhancing overall performance and achieving a more accurate and generalized model that can easily adapt to real-time detection systems.

Figure 5 illustrates the ensemble learning processes of Stacking, Boosting, and Bagging using Random Forests as base models.

## 5. Results and Discussion

In the research environment, we employed Python 3.10.11 on a CPU powered by a 12th Gen Intel Core i7-1260P processor. The study demonstrated that utilizing ensemble learning algorithms, specifically Random Forests and Support Vector Machines, significantly enhanced the performance of intrusion detection models in IoMT systems. By leveraging the strengths of various algorithms through techniques like Stacking and Bagging, the ensemble model mitigated the limitations of individual models and achieved superior predictive accuracy.

Hyperparameter tuning and cross-validation were crucial in optimizing the ensemble model’s performance. Through meticulous adjustment of model parameters and rigorous validation via cross-validation, the study ensured that the ensemble model was robust and adept at handling the diverse and challenging datasets characteristic of IoMT environments. Table 3 provides a brief description of the hyperparameter tuning settings used in the study.

The evaluation metrics utilized in this study, including accuracy, precision, recall, and F1-score, offer valuable insights into the performance of the ensemble learning models applied to classify cyberattacks within IoMT environment. These metrics are pivotal for understanding how well the models perform in real-world scenarios, particularly in critical domains like healthcare, where accurate and timely detection of cyber threats is crucial.

### 5.1. Accuracy

Accuracy is a standard metric for assessing the performance of a classification model. It is calculated as the ratio of correct predictions to the total number of predictions made. In essence, accuracy gauges the fraction of instances that are correctly classified out of all instances [22,35]. The mathematical representation of this equation is as follows:(3)$$\mathrm{Accuracy}=\frac{TP+TN}{TP+TN+FP+FN}$$

### 5.2. Precision

Precision is the proportion of true positive predictions (the number of correctly identified positive instances) relative to the total number of positive predictions made by a model [22,35]. It serves to quantify the accuracy of positive predictions and is computed using the following mathematical concept:(4)$$\mathrm{Precision}=\frac{TP}{TP+FP}$$

### 5.3. Recall

Recall refers to the proportion of true positive predictions (correctly predicted positive instances) out of all actual positive instances in the dataset [22,35]. It serves as a measure of the model’s completeness or sensitivity in identifying positive instances and is computed using the following mathematical concept:(5)$$\mathrm{Recall}=\frac{TP}{TP+FN}$$

### 5.4. F1-Score

The F1-score is a harmonic mean of precision and recall, providing a single metric that balances both measures. It is particularly useful in scenarios where you want to find an optimal balance between precision and recall [22,35]. The mathematical representation of the F1-score is as follows:(6)$$F1=\frac{2\times \mathrm{Precision}\times \mathrm{Recall}}{\mathrm{Precision}+\mathrm{Recall}}=\frac{2\times TP}{2\times TP+FP+FN}$$

The Stacking and Bagging models show remarkable accuracy rates of 98.88% and 97.83%, highlighting their strong potential for use in IoMT environments. Given the sensitivity of medical data and the severe consequences of undetected cyberattacks, these models can provide reliable detection mechanisms essential for maintaining patient safety and healthcare service continuity.

Their near-perfect accuracy indicates a high capability for correctly classifying network traffic, which is crucial in minimizing false negatives. In healthcare, even a single missed threat can have dire implications, making the robustness of these models vital for navigating the complex data landscape of IoMT systems.

In addition to accuracy, the models demonstrate impressive precision and recall rates, ensuring that alerts are likely to be true positives, thereby reducing the risk of unnecessary disruptions. The high F1-scores reflect an effective balance between precision and recall, enhancing their reliability in threat detection.

Both models also exhibit scalability and robustness. Bagging’s approach of aggregating predictions helps manage data variability common in IoMT environments, while Stacking’s use of multiple classifiers allows for adaptability against diverse attack types. Deploying these models could significantly bolster the security infrastructure in healthcare, ensuring the integrity and confidentiality of patient information while maintaining high performance in dynamic conditions.

The Boosting technique exhibited a lower accuracy rate of 88.68%. Several factors contribute to the comparatively lower accuracy of Boosting models in relation to other methods. One primary reason for the challenges faced by Boosting methods, such as Gradient Boosting or AdaBoost, lies in their sequential approach to error correction from preceding models. This iterative process can result in a specialized model that may overfit to the training data, particularly when the dataset contains noise or outliers. The sequential nature of Boosting, which prioritizes learning from past errors, can also introduce greater model complexity and longer training durations, potentially reducing robustness to data variations compared to ensemble methods that aggregate predictions.

Table 4 presents the outcomes of ensemble learning models, including Stacking, Bagging, and Boosting, in terms of accuracy, precision, recall, and F1-Score. Additionally, it provides a comprehensive overview of the training and prediction times per sample for each model, offering insights into their performance and efficiency.

Table 5 presents a comparative analysis between the proposed scheme and recent studies that employed the same dataset. Our models demonstrate a promising performance according to the evaluation metrics. Consequently, the experimental results confirm that the proposed scheme effectively classifies cyberattacks in the IoMT.

To assess the statistical significance of the difference in accuracy between our model and other models, a Z-Test for proportions was utilized.

For Comparison with Study 1 using pooled proportion (96.90%):$$p_{\mathrm{pooled}\_1}=\frac{\left(0.9888\times 5709)+(0.9690\times 5709\right)}{5709+5709}=\frac{5645.59+5527.92}{11418}=0.97885$$

For Comparison with Study 2 using pooled proportion (94.50%):$$p_{\mathrm{pooled}\_2}=\frac{\left(0.9888\times 5709)+(0.9450\times 5709\right)}{5709+5709}=\frac{5645.59+5399.01}{11418}=0.9663$$

Then, we computed the standard error for each comparison.

Study 1:$$SE_{1}=\sqrt{0.97885\times \left(1-0.97885\right)\times \left(\frac{1}{5709}+\frac{1}{5709}\right)}\approx 0.0026$$

Study 2:$$SE_{2}=\sqrt{0.9663\times \left(1-0.9663\right)\times \left(\frac{1}{5709}+\frac{1}{5709}\right)}\approx 0.0036$$

Finally, we calculated the Z-score for each comparison.

Study 1:$$Z_{1}=\frac{0.9888-0.9690}{0.0026}=\frac{0.0198}{0.0026}\approx 7.62$$

Study 2:$$Z_{2}=\frac{0.9888-0.9450}{0.0036}=\frac{0.0438}{0.0036}\approx 12.17$$

A Z-score of 7.62 and 12.17 corresponds to *p*-values that are far less than 0.05, indicating highly statistically significant differences.

The improvements in accuracy from 96.90% to 98.88% and from 94.50% to 98.88% are statistically significant. This means the enhancement in performance is not due to random chance but represents a real and meaningful improvement in the model’s capability. This conclusion strengthens the validity of the proposed method’s performance compared to previous studies.

Figure 6, Figure 7 and Figure 8 present the confusion matrix, providing a clear summary of each model’s classification performance by displaying the number of correct and incorrect predictions.

Figure 9 represents the Performance Visualization of ROC curves for the Stacking, Bagging, and Boosting models. The ROC curve (Receiver Operating Characteristic curve) is a graphical representation of a classification model’s performance across various threshold settings. It illustrates the trade-off between sensitivity (true positive rate) and 1-specificity (false positive rate). The Area Under the Curve (AUC) quantifies the model’s overall ability to distinguish between positive and negative classes; a higher AUC indicates superior performance.

For the Stacking model, the AUC is reported as 1.00, which signifies a perfect classification performance. This means that the Stacking model achieves flawless separation between positive and negative classes across all threshold levels, classifying all positive cases correctly and all negative cases correctly. This exceptional performance is reflected in its accuracy rate of 98.88%, demonstrating its effectiveness in correctly identifying the class of each sample.

The Bagging model, with an AUC of 0.99, also performs extremely well, showing a slight decrease in performance compared to Stacking. An AUC of 0.99 indicates that the Bagging model nearly achieves perfect separation between classes. This high AUC corresponds with its accuracy rate of 97.83%, which confirms its high level of classification accuracy and reliability.

In contrast, the Boosting model exhibits an AUC of 0.89, indicating good but less optimal performance compared to the other models. While still demonstrating strong classification ability, the lower AUC reflects a slight reduction in its ability to distinguish between classes. This performance is consistent with its accuracy rate of 88.68%, which, while still high, is lower than that of the Stacking and Bagging models.

The “Random Guess” line, depicted as a diagonal from the bottom-left to the top-right of the ROC plot, represents the performance of a model that makes predictions randomly. It serves as a baseline; any model with an ROC curve above this line performs better than randomly guessing, showcasing the model’s ability to discern meaningful patterns in the data.

While the models exhibit high-performance metrics such as accuracy, precision, recall, and F1 score, their true applicability to real-world scenarios depends on factors beyond these measures. For instance, in practical settings, the models’ effectiveness is influenced by the quality and quantity of data, the balance between different classes, and the presence of unseen or novel patterns in the data. Additionally, real-world deployment involves considerations such as computational resources, interpretability of results, and adaptability to changing conditions. Therefore, despite the strong theoretical performance indicated by the reported metrics, further investigation is necessary to evaluate how well these models perform in diverse and dynamic real-world environments.

### 5.5. Limitation

One significant limitation of employing ensemble learning techniques for intrusion detection is the propensity for overfitting. The iterative nature of most machine learning algorithms, where errors from preceding models are sequentially corrected, can inadvertently lead to the creation of highly specialized models that excessively tailor themselves to the specifics of the training data. This overfitting phenomenon becomes particularly pronounced when the dataset contains noisy or outlier-laden data points, potentially compromising the model’s ability to generalize well to unseen instances. Furthermore, the inherent complexity introduced by ensemble learning methods and the associated longer training durations may present challenges in adapting to diverse data patterns and variations, ultimately diminishing the overall robustness of the IDS.

Additionally, the availability of comprehensive IoMT datasets poses a limitation. Often, such datasets are either not publicly accessible or lack the diversity required to train robust machine learning models. This scarcity of high-quality, representative data can hinder the development and evaluation of effective IDS solutions as models may not fully capture the spectrum of potential intrusion scenarios. The limited availability of IoMT datasets also means that models trained on these datasets might struggle to perform effectively in real-world environments, where the data characteristics can differ significantly from the training data. This underscores the need for more extensive and diverse IoMT datasets to enhance the reliability and generalizability of IDS models.

### 5.6. Future Work

In charting the trajectory for future research endeavors in intrusion detection through ensemble learning methodologies, a promising avenue for exploration lies in devising strategies to counteract the overfitting tendencies of Boosting algorithms. Researchers can delve into the development and implementation of regularization techniques or ensemble pruning methods to curb overfitting and enhance the generalization capabilities of Boosting models. Exploring the nuanced interplay between model complexity and performance in Boosting algorithms represents another fertile area for investigation, aiming to strike an optimal balance that ensures both efficiency and adaptability across varying data scenarios.

Moreover, the exploration of novel ensemble learning paradigms that seamlessly integrate the strengths of Boosting algorithms with sophisticated regularization mechanisms holds the potential to yield more resilient and precise intrusion detection models tailored specifically for safeguarding sensitive data within IoMT environments. By advancing the understanding of how to effectively manage overfitting challenges in Boosting methods, researchers can pave the way for the development of cutting-edge IDS that excel in accuracy, reliability, and adaptiveness within the complex landscape of healthcare cybersecurity.

## 6. Conclusions

This research offers a comprehensive evaluation of the effectiveness of machine learning models for intrusion detection within the Internet of Medical Things (IoMT), aiming to fortify cybersecurity measures and protect sensitive healthcare data. The study primarily focused on ensemble learning techniques—Stacking, Bagging, and Boosting—using Random Forests and Support Vector Machines (SVM) as base models, and evaluated these approaches on the WUSTL-EHMS-2020 dataset.

The analysis of performance metrics, including accuracy, precision, recall, and F1-score, highlights that Stacking achieved exceptional accuracy and reliability, with an accuracy rate of 98.88%. Bagging followed closely with an accuracy rate of 97.83%, while Boosting, although effective, recorded a lower accuracy rate of 88.68%. These results underscore the superior performance of Stacking and Bagging in detecting and classifying cyber attack incidents, showcasing their potential to significantly enhance IoMT security.

Despite these promising results, deploying these models in real-world medical environments poses several challenges. The computational demands and real-time data processing requirements are critical for ensuring effective operation in dynamic healthcare settings. Additionally, integrating these models with existing healthcare systems and addressing compatibility issues are crucial for their successful implementation. The study’s reliance on a single experimental dataset limits the generalizability of the findings, suggesting a need for further validation across diverse datasets and real-world scenarios.

Future research should focus on addressing these practical challenges, particularly by exploring the adaptability of these models to various data variations and evolving threat landscapes. Investigating the integration of these models into existing healthcare infrastructure, improving computational efficiency, and ensuring robust real-time performance are essential steps towards realizing their full potential.

The insights gained from this study are pivotal for developing robust IoMT security solutions, ensuring that the numerous benefits of IoMT—such as improved patient outcomes and personalized care—are safeguarded against cybersecurity threats. By advancing machine learning techniques within intrusion detection systems, healthcare providers can enhance their capability to monitor and protect their networks, thus maintaining the confidentiality, integrity, and availability of critical healthcare information. Future work should continue to address the challenges of real-world deployment and refine these models to ensure their effectiveness in a wide range of practical applications.

## Figures and Tables

**Figure 1 sensors-24-05937-f001:**
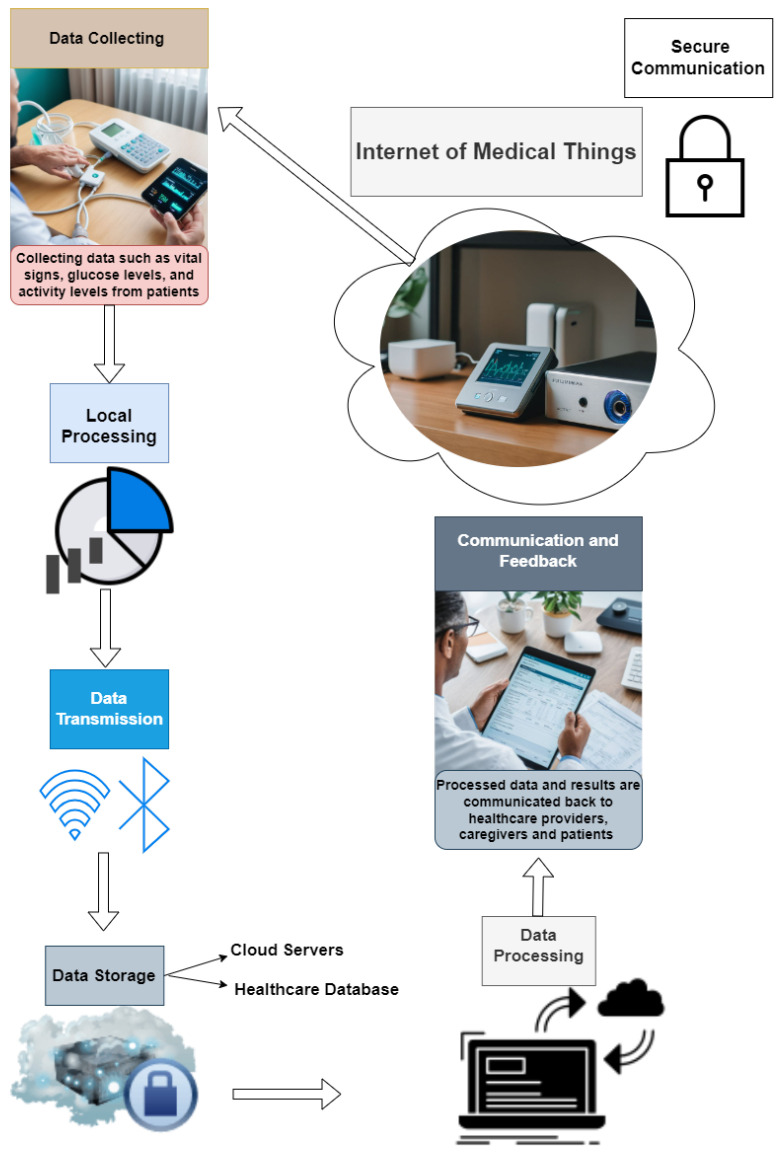
This figure shows the data flow of the IoMT.

**Figure 2 sensors-24-05937-f002:**
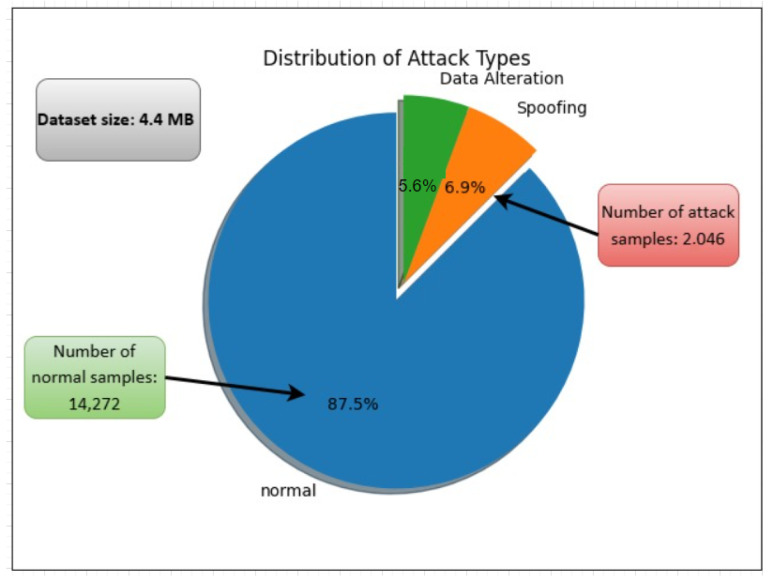
Summary of the dataset’s statistical attributes.

**Figure 3 sensors-24-05937-f003:**
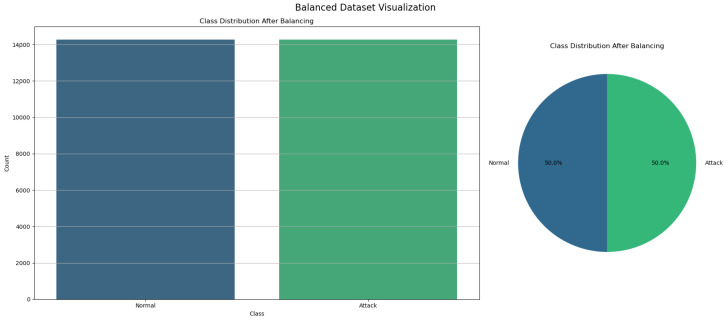
Balanced dataset after random over-sampling.

**Figure 4 sensors-24-05937-f004:**
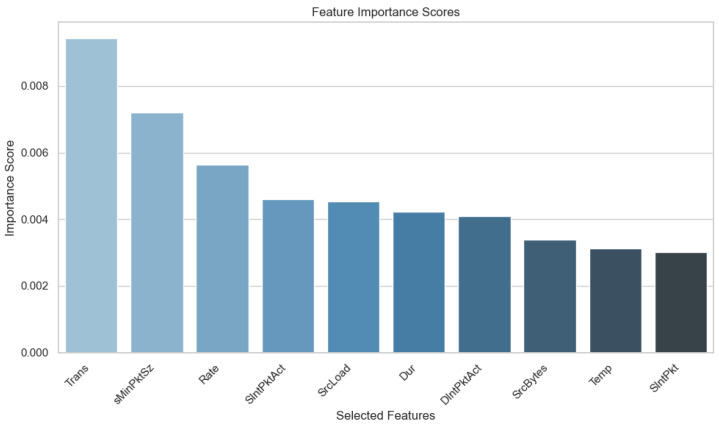
Top 10 features selected by mutual information.

**Figure 5 sensors-24-05937-f005:**
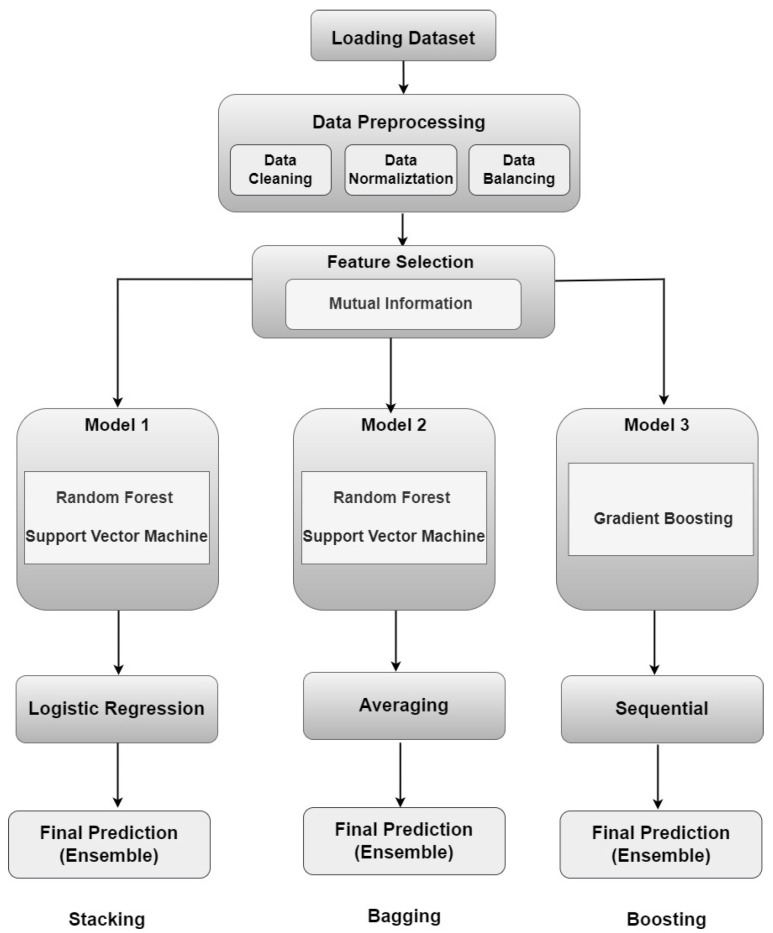
The processes of stacking, boosting, and bagging ensemble learning techniques.

**Figure 6 sensors-24-05937-f006:**
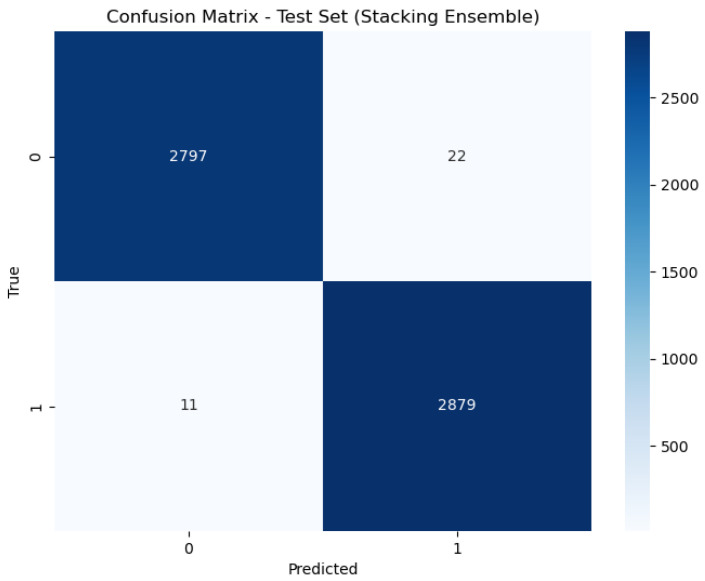
Confusion matrix of stacking.

**Figure 7 sensors-24-05937-f007:**
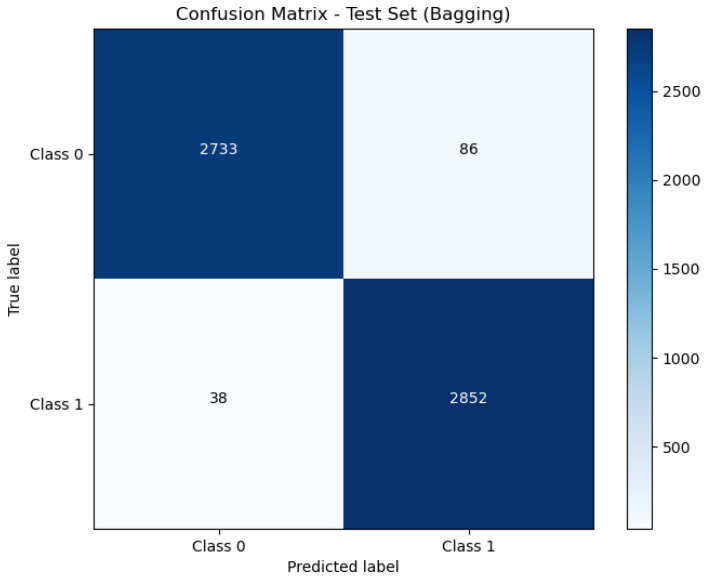
Confusion matrix of Bagging.

**Figure 8 sensors-24-05937-f008:**
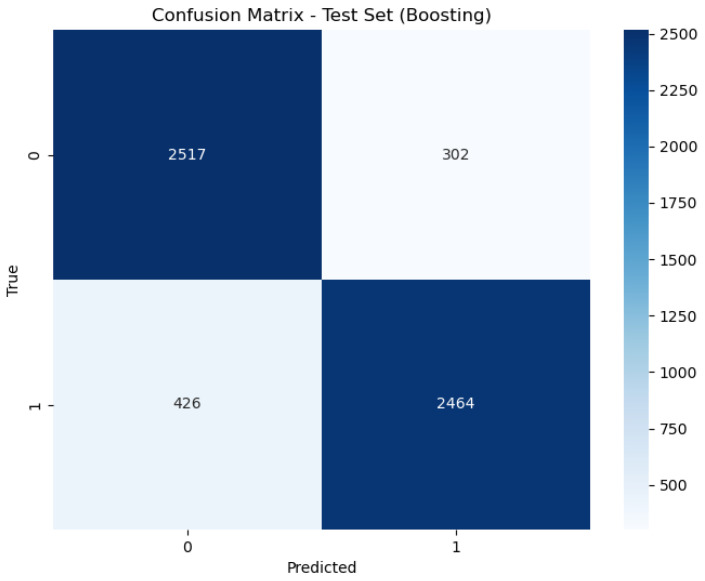
Confusion matrix of Boosting.

**Figure 9 sensors-24-05937-f009:**
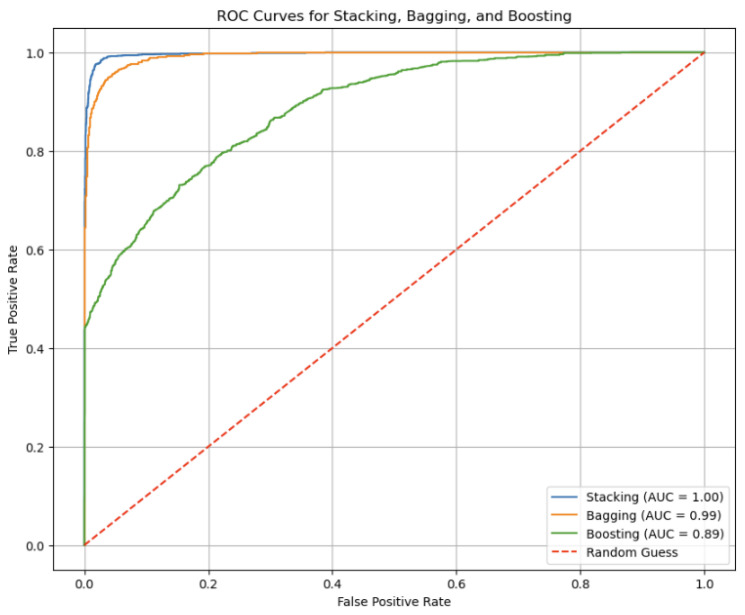
ROC curve of Boosting.

**Table 1 sensors-24-05937-t001:** Comparison of our scheme with recent studies.

Reference	Year	Dataset	Type of Attacks	Method	Accuracy
[21]	2022	WUSTL EHMS 2020	Spoofing Data injection	Ensemble Learning, Stacking	96.9%
[22]	2023	TON-IoT	Backdoor, DDoS, Injection, Password, Ransomware, Scanning, XSS	Ensemble Learning, Stacking Voting	98.64% 96.63%
[23]	2023	WUSTL EHMS 2020	Spoofing Data injection	Ensemble Learning, Stacking	94.50%
[24]	2023	WUSTL EHMS 2020	Spoofing Data injection	Ensemble Learning	96.56%
[25]	2022	WUSTL EHMS 2020	Spoofing Data injection	Ensemble Learning	94.23%
[26]	2024	WUSTL EHMS 2020	Spoofing Data injection	Federated Bayesian Optimization XGBoost	96.00%
Our Research	2024	WUSTL EHMS 2020	Spoofing Data injection	Ensemble Learning, Stacking Bagging Boosting	98.88% 97.83% 88.68%

**Table 2 sensors-24-05937-t002:** Dataset features description.

Metric	Description	Type
Trans	Aggregated packets count	Flow metric
sMinPktSz	Source minimum transmitted packet size	Flow metric
Rate	Number of packets per second	Flow metric
SIntPktAct	Source active inter packet arrival time	Flow metric
SrcLoad	Source load (bits per second)	Flow metric
Dur	Duration	Flow metric
DIntPktAct	Destination active inter packet arrival time	Flow metric
SrcBytes	Source bytes in the flow record	Flow metric
Temp	Temperature	Biometric
SIntPkt	Source inter packet arrival time	Flow metric

**Table 3 sensors-24-05937-t003:** Summary of the optimized hyperparameters.

Algorithm	Hyperparameter
Stacking	param_grid_stacking = { ’rf_dist’: { ’n_estimators’: randint(50, 300), ’max_depth’: [None, 10, 20, 30], ’min_samples_split’: randint(2, 10), ’min_samples_leaf’: randint(1, 4), ’max_features’: [’auto’, ’sqrt’, ’log2’] } }
Bagging	param_grid_bagging = { ’rf_dist’: { ’n_estimators’: randint(50, 300), ’max_depth’: [None, 10, 20, 30], ’min_samples_split’: randint(2, 10), ’min_samples_leaf’: randint(1, 4), ’max_features’: [’auto’, ’sqrt’, ’log2’] } }
Boosting	param_grid_boosting = { ’n_estimators’: randint(50, 300), ’learning_rate’: uniform(0.01, 0.5), ’max_depth’: randint(3, 15), ’min_samples_split’: randint(2, 10), ’min_samples_leaf’: randint(1, 5) }

**Table 4 sensors-24-05937-t004:** Evaluation results of the ensemble models.

Algorithm	Accuracy	Precision	Recall	F1-Score	Training Time (per Sample)	Prediction Time (per Sample)
Stacking	98.88%	98.23%	99.58%	98.90%	0.011487 s	0.000010 s
Bagging	97.83%	97.07%	98.69%	97.87%	0.001558 s	0.000161 s
Boosting	88.68%	88.00%	88.80%	88.57%	0.001582 s	0.000310 s

**Table 5 sensors-24-05937-t005:** Our proposed method was compared to recent studies that utilized the same dataset.

Model	Accuracy	Precision	Recall	F1-Score
[21]-Stacking	96.90%	96.50%	96.00%	96.00%
[23]-Stacking	94.50%	94.50%	94.50%	94.50%
Proposed Method Stacking	98.88%	98.23%	99.58%	98.90%

## Data Availability

Data are available upon request.

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
