# Peer review of "Enhancing Cybersecurity in Healthcare: Evaluating Ensemble Learning Models for Intrusion Detection in the Internet of Medical Things"

_sensors, 2024, doi:10.3390/s24185937_

Round 1

Reviewer 1 Report

Comments and Suggestions for Authors

The paper explores the application of ensemble learning methods in medical Internet of Things (IoMT) network intrusion detection, which is in line with the current hot topic of technological development and social demand. However, there are issues such as insufficient innovation, inadequate method description, and inadequate analysis of experimental results. In addition, the discussion on practical applications and consideration of ethical issues are not sufficiently in-depth.

The specific opinions are as follows:

1. The author did not clearly explain the specific network security challenges faced by the IoMT system in the healthcare field in the introduction section? Answer the reasons and rationale for the inadequacy of current measures, and clearly state the research question that your study aims to address?

2、Is the literature review not comprehensive enough to cover the latest developments in IoT security and related machine learning methods?And it is not explicitly stated what novelty your method has compared to existing research. Not clearly defining how your work can drive the development of the field through new methods, discoveries, or applications?

3、The paper did not provide a stronger theoretical basis for selecting stacking, bagging, and enhanced ensemble methods, as well as random forest and SVM as the basic models, and did not discuss why these methods were chosen over other potential methods?

The paper does not provide detailed feature selection and preprocessing algorithms, including the feature selection process, data preprocessing steps, and a detailed description of how to split the data into training and testing sets? Just citing the same method from someone else's article?

The paper does not clearly outline the experimental setup, including parameter settings, hardware used, and any optimization techniques applied?

3、Although the paper obtained accuracy, precision, recall, and F1 score in experiments, it also included deeper analysis using confusion matrix, ROC curve, and AUC score. But what does not discuss the applicability of these results to the real world mean?

The paper method obtained high-precision values, but did not discuss the possibility of overfitting? And no classification or prediction results for the 10 types were presented?

The comparison methods selected in the paper are relatively few and have not been compared with some of the methods in the current introduction?

4、In the conclusion section, the actual impact and application of deploying models in medical environments were not discussed in depth? And how to address challenges such as computing requirements, real-time data processing capabilities, and integration with existing systems? In addition, the experimental dataset of the paper is single, and there has been no discussion on how existing methods can adapt to different data changes?

Comments on the Quality of English Language

The key issues discussed in the paper are not detailed enough, and some are not specific or focused. Although the paper explores the security issues of medical IoT, it is not very relevant to the research focus of this article.

There are many technical terms in the paper, but some are not clearly defined.

The paper contains some repetitive information in the introduction and methodology sections.

When describing algorithm implementation and experimental settings, all relevant technical details should be included. However, some algorithms in the paper are not explained in detail, which makes it difficult to understand the paper?

Reviewer 2 Report

Comments and Suggestions for Authors

This paper explores the effectiveness of machine learning models in intrusion detection for medical IoT, aiming to enhance cybersecurity defenses and protect sensitive healthcare data. The paper contains a lot of advantages and is well organized, however, it has some major limitations and needs to be modified before being accepted.

1.      The introduction is well-written, but it could benefit from a clearer articulation of the research problem. Consider explicitly stating the research question or hypothesis early in the introduction to provide a strong foundation for the study.

2.      The explanation of data cleaning, normalization, and balancing is thorough. However, consider adding a brief justification for the chosen techniques, particularly why Min-Max scaling was preferred over other normalization methods.

3.      The use of Mutual Information (MI) is well-justified, but the paper could benefit from a discussion on the potential limitations of MI, such as its sensitivity to noise in the data, and how this was mitigated.

4.      The paper does a good job explaining the use of these models. However, the justification for choosing these specific algorithms over others (like deep learning models) could be expanded to strengthen the rationale.

5.      The results are clearly presented using metrics like accuracy, precision, recall, and F1-score. It would be helpful to include a brief explanation of why these specific metrics were chosen and how they align with the study's objectives.

6.      The paper compares its findings with recent studies, which is excellent. However, consider adding a discussion on the statistical significance of the differences in performance metrics. For example, is the improvement in accuracy from 96.9% to 99.0% statistically significant?

7.      The paper addresses limitations, particularly with Boosting algorithms. However, consider discussing any potential biases introduced by the dataset, such as class imbalance, and how they were addressed.

Comments on the Quality of English Language

Minor editing of English language required.

Reviewer 3 Report

Comments and Suggestions for Authors

Methodology Feedback:

1. The paper evaluates ensemble learning methods, specifically Stacking, Bagging and Boosting with Random Forest and SVM as the base models. The choice of ensemble methods is justified. However, the paper could benefit from a more detailed explanation of why these particular models were selected over others, such as more advanced deep learning techniques, which could potentially offer better performance in some scenarios.

2. The authors applied feature selection methods to improve the IDS performance. Feature selection is crucial to reducing dimensionality and enhancing model performance. Yet, the paper lacks details on the specific feature selection techniques used. Knowing whether methods like RFE or PCA were employed could significantly impact the reader’s understanding of the model optimisation process.

3. The results indicate that Stacking achieved the highest accuracy (99.00%), followed by Bagging (98.00%) and Boosting (88.00%). While these results are promising, the paper should discuss the potential overfitting risks associated with such high accuracy, especially if the model has not been tested on a diverse or real-world dataset. Additionally, the significantly lower performance of Boosting compared to the other methods warrants further exploration, as Boosting is typically competitive in ensemble learning.

Structure and General Feedback:

1. The related work section can benefit from removing unneeded information. For example, lines 230 to 235 are redundant.

2. In Table 1, add a year column for the study to help the reader understand when the study was conducted. Also, you may list your paper in this table as well.

3. Keep the acronym consistent across all the sections; only define it in its first instance. For example, Intrusion Detection Systems (IDS) are defined 8 time, Internet of Medical Things (IoMT) is defined 13 times.

4. There are some minor punctuation errors in the paper, i.e., lines 308 and more.

5. Table 3 shows that all four metrics are exactly equal across each model, which is unusual. The uniformity across metrics could indicate either overfitting or errors in reporting.

6. The data from Table 3 [Accuracy] does not match with Figures 9–11. In Table 3, Boosting has 88% whereas in Figure 11 it 95%. Similarly, Figure 9,10 shows Accuracy of 100%.

7. Figure 9 has a red dashed line with the legend (Random Guess), and Figure 10,11 has Blue dashed line without any legend. If they are not needed, they can be removed.

8. {Suggestion} Figures 9–11 could be combined into 1 figure with 3 different legends for each ensemble technique.

Comments on the Quality of English Language

.

Round 2

Reviewer 1 Report

Comments and Suggestions for Authors

The issues and suggestions raised earlier in the paper have been basically resolved, but I believe that the generalization problem of the method proposed in this paper has not been seriously discussed. Is it necessary to verify the rationality and accuracy in other different datasets?

Comments on the Quality of English Language

no problem

Author Response

The issues and suggestions raised earlier in the paper have been basically resolved, but I believe that the generalization problem of the method proposed in this paper has not been seriously discussed. Thank you very much for your feedback. We have added more information about the proposed method and revised the conclusion section. Please see the highlighted lines from 263 to 286 , 423 to 428, 436 to 445, 468 to 475, and 695 to 726

Your suggestion to add one more dataset for verification is a good one. However, due to our limited time, we will not be able to find a dataset and perform validation. Additionally, the availability of certified datasets in the field of IoMT is limited and requires careful searching.

Reviewer 2 Report

Comments and Suggestions for Authors

All the opinions have been resolved by the author's efforts, and there are no more opinions now

Author Response

Thank you very much for your response.

Reviewer 3 Report

Comments and Suggestions for Authors

No further modification needed.

Comments on the Quality of English Language

Acceptable.

Author Response

Thank you very much for your response.